# Incidence, healthcare-seeking behavior and barriers associated with seeking care for severe childhood pneumonia in rural Bangladesh: A prospective study

**Shahjahan Ali**[1], **Md. Tariqujjaman**[1]*, **Arifa Farzana Tanha**[1], **Marufa Sultana**[1,2], **Sabiha Nasrin**[1], **Kamal Ibn Amin Chowdhury**[1], **Mohammod Jobayer Chisti**[1], **Nur H. Alam**[1], **Tahmeed Ahmed**[1], **Niklaus Gyr**[3], **Abu S. G. Faruque**[1], **George J. Fuchs**[4]

1 Nutrition Research Division, icddr,b, Dhaka, Bangladesh, 2 Faculty of Health, Deakin Health Economics, Institute of Health Transformation, School of Health and Social Development, Deakin University, Melbourne, Victoria, Australia, 3 Faculty of Medicine, University of Basel, Basel, Switzerland, 4 Department of Pediatrics, University of Kentucky College of Medicine and Kentucky Children's Hospital, Lexington, Kentucky, United States of America

☯ Equal contribution
* md.tariqujjaman@icddrb.org

## Abstract

Globally, childhood pneumonia is one of the leading causes of under-five mortality especially in low-and middle-income countries. This study aimed to estimate the incidence, healthcare-seeking behavior, and barriers associated with seeking care for children suffering from severe pneumonia in rural Bangladesh. A prospective study was conducted in two districts at 81 randomly selected villages in rural Bangladesh. We collected data on 12,303 under-five children between 1st June 2018 to October 2019. Among surveyed children, a total of 154 severe pneumonia cases were recorded, resulting in an overall incidence of 1.3%. When seeking healthcare for their children, most (44.2%) mothers/caregivers availed the health care services from nearby pharmacies or local healthcare providers, followed by Health and Family Welfare Centre (37.0%), private clinics (23.4%), district hospitals (23.4%) health complexes (22.1%). Only 2% sought care at community clinics. Approximately 25% (n=38) of caregivers reported barriers to accessing healthcare. Among those who reported barriers, 39.5% cited an inability to find transportation as the main issue, followed by 26.3% who mentioned high cost of treatment, 10.5% who faced natural calamities including floods, and 2.6% who encountered political instability including strike. The incidence of severe pneumonia was below 2% in our rural areas of Bangladesh. Pharmacies and local healthcare providers were the most commonly used sources for care-seeking. Notable barriers to care-seeking included the lack of available transportation and the high cost of treatment.

## Introduction

More than 700,000 children under five years of die from pneumonia each year, which is estimated to be 2,000 every day, making pneumonia the infectious disease that claims the

**Data availability statement:** Data access is restricted for open source since it was primary data from a study conducted by icddr,b and the funder institutions of this project have restrictions for public access to this data. However, data will be provided to interested researchers from the icddr,b Data Repository (Research Administration's Data Repository) for purposes of secondary data analyses upon approval of a Data Licensing Application & Agreement (Application) by the icddr,b Data Repository Committee. Interested personnel are recommended to consult this with icddr,b's IRB Coordinator Md. Abdus Salam Khan (salamk@icddrb.org).

**Funding:** This study was funded by Kentucky Children's Hospital. The funders had no role in study design, data collection and analysis, decision to publish, or preparation of the manuscript.

**Competing interests:** The authors have declared that no competing interests exist.

majority of lives in children. The highest incidence is found in South Asia, where there are 2,500 cases per 100,000 children, and in West and Central Africa, where there are 1,620 cases per 100,000 children [1]. Moreover, outside of the neonatal period, pneumonia remains the most common cause of morbidity for young children, especially in low- and middle-income countries (LMICs) [2]. Despite, worldwide initiatives to expand the availability of the life-saving pneumococcal conjugate vaccine, pneumonia continues to be the primary cause of mortality and morbidity in children under five [3]. Reports indicate that the number of clinical pneumonia cases in young children worldwide decreased by 22% between 2010 and 2015 [4]. Despite the global reduction, pneumonia was still responsible for 15% of all deaths of children aged less than five years in Bangladesh in 2015, out of 119,000 total cases [5].

Pneumonia is an acute respiratory infection of the lungs. Children with pneumonia experience short, painful breathing, low oxygen levels, and possibly even death due to the buildup of purulence and fluid in the lung alveoli due to lung inflammation [6]. The disease may worsen due to immune system impairment, deformities of the chest wall, soft rib anomalies, and muscle weakness [7–9]. Pneumonia-associated mortality in young children is attributed to several factors including cyanosis [10], inability to feed orally, malnutrition [10,11], prolonged illness duration [12], altered psychological state [11], as well as the existence of underlying chronic diseases (such as heart disease) [13]. A majority of pneumonia cases (>98%) and pneumonia-related deaths (>99%) occur in LMICs and primarily during the first year of life and in the community outside of hospital setting [14]. In LMICs, 48 million deaths (seven million of which were child deaths) occurred in 2010; the majority in rural communities with limited access to healthcare [15,16]. Child mortality rates are recognized by most to be higher in rural than urban areas [17]. Parents in rural areas are less aware of the clinical signs and symptoms of pneumonia and or to consider the illness to be serious or life-threatening [18].

Deaths due to pneumonia stayed consistent before and after the Hib vaccine was introduced [19]. In Bangladesh among under-five children, pneumonia-related mortality accounts for 9,511 deaths per 100,000 individuals annually [20]. A case fatality rate of 10% was reported among 47,228 children hospitalized for severe or very severe pneumonia in a study assessing the pneumonia-specific case fatality rate over time after a Child Lung Health Program (CHLP) implementation within Malawi's national health systems [21]. Another cross-sectional study in Bangladesh revealed that access to timely care for childhood pneumonia is associated with several factors including low levels of education, financial difficulties, poor transportation system, and distance to medical facilities [22]. To address the primary causes of child mortality, epidemiologists and social scientists now place a higher priority on definitions of health-seeking behavior [23]. Seeking the right medical attention at the right time can reduce under-five mortality from severe pneumonia [18,24,25]. In addition to obtaining data from health facilities, such as health services delivery, it is essential to comprehend parents' views and the barriers to their behavior when seeking medical attention for their child's pneumonia in order to inform healthcare policies and programs to enhance health care utilization in underprivileged areas [26,27].

To improve the health facility services and coverages and address barriers to timely healthcare seeking, the government of Bangladesh has placed importance on research and evidence-based interventions. Some cases may be treated at home or may not receive any treatment at all due to inadequate care-seeking behavior. Thus, home visits are needed to gather information about children with pneumonia, determinants of parents/caregivers medical attention seeking, and the most critical factors leading to child death. At present, the majority of the literature currently accessible in Bangladesh and other LMICs is from studies in hospital settings. Such studies were performed over relatively short observation periods of time and/or with small sample sizes in constrained geographic regions. Additionally, we could not identify any research that were representative of countries with heterogeneity

in socioeconomic demographics and healthcare-seeking practices. It remains essential to identify severe pneumonia rates, associated characteristics of severe pneumonia, and current treatment-seeking practices to develop practical and evidence-based interventions to improve healthcare-seeking behavior for severe childhood pneumonia and reduce the burden of severe childhood pneumonia morbidity and mortality. The aim of the current study was to define the incidence of severe pneumonia, care-seeking behavior, and the barriers of care-seeking behavior among mothers and caregivers in rural Bangladesh. Such evidence is essential to formulate appropriate and effective strategies to improve timely healthcare-seeking behavior for children suffering with severe pneumonia.

## Methods and materials

### Study design

The epidemiological survey was nested in an intervention study of children with severe pneumonia with or without malnutrition in the health system of Bangladesh [28]. The primary study was designed as a prospective, cluster-randomized controlled clinical trial (registered at www. ClinicalTrials.gov:NCT02669654). This trial aimed to define the clinical and cost-effectiveness of a Day Care Approach alternative to Usual Care (comparison group) within the Bangladesh health system to manage severe childhood pneumonia. The study was approved by the Institutional Review Board (IRB) of icddr,b comprising the Research Review Committee (RRC) and Ethical Review Committee (ERC). As the population-based incidence of severe pneumonia is of particular interest to this epidemiological study, we determined the total number of children with severe pneumonia who reported to Health and Family Welfare Centers (HFWCs), sub-district, and district facilities. The number of severe childhood pneumonia cases from those who sought care from other sources of service delivery such as different levels of health care from health care providers of the locality and private facilities or clinics were also included to enhance the number of respondents and capture maximum information as much as possible.

### Participants

To estimate the population-based incidence of severe pneumonia cases, we first calculated the number of severe pneumonia children reported to each of the predefined HFWCs and sub-district and district facilities from the study catchment areas (unions with an average population size of 25,000). Then, we adjusted the childhood severe pneumonia data from the community by adding the number of severe childhood pneumonia cases (numerator) that developed in the catchment areas but did not report to HFWCs, sub-district hospitals, and district hospitals but reported to other sources of health care for severe pneumonia episodes from the catchment areas (denominator: a population at risk). Children with severe pneumonia and caregivers were eligible for the current study.

### Sample size calculation

The study captured significant variability because the information was collected from numerous households in both clusters (unions) across all study sites (sub-districts). The sample size was calculated using the formula, $n = \dfrac{Z_\alpha^{\;2}\;P(1-P)}{d^2}$ , where p is the incidence of severe pneumonia in under-five children (2.02%), $Z_\alpha$ is 1.96 at a 5% level of significance, and d is the desired precision (0.5%). Using these parameters, the minimum sample size was determined to be 3042. Considering a design effect of 3 and a 10% non-response rate, the total sample size required was 10040 children under 5 years old. The study also tracked severe pneumonia

episodes over 12 months. It was assumed that during this period, about 80% of severe pneumonia cases would be captured from HFWCs, sub-district hospitals, and district hospitals, as distance is a significant factor in seeking health care. The survey estimated the number of new cases by collecting information using both prospective and retrospective approaches, with the population size serving as the denominator in the past year.

## Study setting and sampling procedure

This study was conducted in Dhaka and Kishoregonj districts in Bangladesh. From the Dhaka district, we selected the Dhamrai sub-district, and from the Kishoreganj district, we selected the Karimganj, Pakundia, and Kishoreganj Sadar sub-districts for data collection (Fig 1). We followed a multi-stage sampling technique for collecting data. From each of the 32 unions, we interviewed 314 households having at least one under-five child. A total of 10,048 households were interviewed. The survey team randomly prepared a list of 4 potential villages (2 villages having less than 100 households and another two villages with more than 100 households) for each union by computer-generated numbers for this survey. To begin with, in half of the unions (n=16), the survey was started from a randomly selected small village; once all households of that small village were covered, the survey team worked in a randomly selected large village until the sample size of 314 in that union was reached. In the rest of the unions (n=16), the same procedure has been applied. If the village has more than 314 households, then activities were limited to that village only. The children were identified by the door-to-door household visits having at least one under-five children in the households.

## Data collection instruments and variables

The data collection team visited households to identify severe pneumonia cases in the pre-selected clusters within the catchment areas of HFWCs (both existing and newly expanded areas: 16+16=32 catchment areas). Team members first identified severe pneumonia cases, and then administered a questionnaire to collect information on symptoms of illness, health care utilization, and demographic characteristics. Two questionnaires, i.e., a household questionnaire and a separate questionnaire for under-5 children, were administered to collect data. Initial questionnaires were pre-tested in the field and circulated to relevant professionals for feedback. Both questionnaires were finalized after conducting a series of meetings with internal investigators and translated and printed in Bengali. The household questionnaire consisted of a schedule for listing all household members. The survey collected relevant information for each listed person, such as age, sex, marital status, education, and occupation. In addition, data on the type of housing, water sources, sanitation, availability of electricity, and ownership of household assets were collected. The under-5 questionnaire included questions on morbidity, mortality, and healthcare-seeking behavior including access to healthcare services and types of barriers for care-seeking.

## Statistical analysis

Data management including data entry and data cleaning were performed using Statistical Package for Social Sciences (SPSS), version 20.0 (Windows SPSS, Chicago, IL). Data analysis was performed by using Stata 15 software. The study sites map was developed using R studio (v 4.4.2). The outcome of interest is the incidence of severe pneumonia and the healthcare-seeking behavior. Pneumonia was defined as a history of cough or difficulty breathing, and lower chest wall in-drawing or age-specific rapid breathing (≥50 breaths per minute for 2–11 month-olds and ≥40 breaths per minute for 12–59 month-olds), without any general danger signs [29,30]. Severe pneumonia was defined as per World Health Organization (WHO)

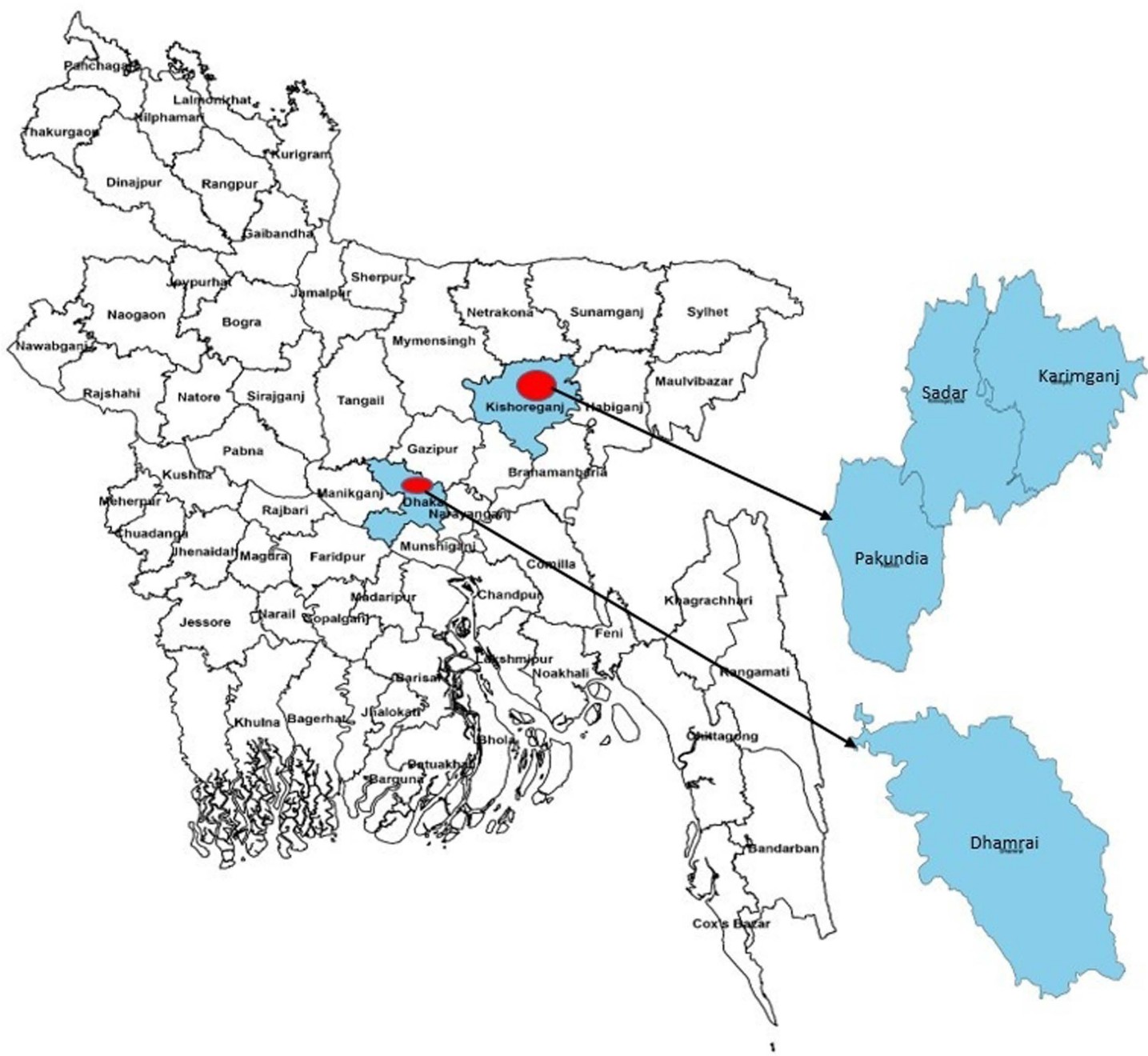

**Fig 1. Study sites.** The district and sub-district-level shapefiles of Bangladesh were obtained from the open-source website Humanitarian Data Exchange (https://data.humdata.org/dataset/cod-ab-bgd).

guidelines as pneumonia with at least one of the following danger signs: central cyanosis or hypoxemia (oxygen saturation < 90% measured by pulse oximeter), severe respiratory distress (including grunting, very severe chest in-drawing), inability to breastfeed or drink, lethargy or unconsciousness, or convulsions [30]. The Incidence of childhood severe pneumonia was calculated by the new cases of severe pneumonia in the household by the total number of households surveyed. Then the incidence was multiplied by 100 to get the percentage of severe

pneumonia incidence. Health care-seeking behavior was defined by place for availing health care services. The socioeconomic, clinical, and demographic characteristics of severe pneumonia children were presented in frequencies with percentages. The healthcare-seeking behavior including access to healthcare services, and barriers to receiving healthcare services was presented in frequencies with respective percentages. The types of drugs received were presented in a bar chart. The presence of barriers faced by caregivers was illustrated in a pie chart, while the types of barriers were presented in a bar chart.

## Ethics approval and consent to participate

The protocol of this study was reviewed by the IRB of icddr,b which consisted of two committees: the RRC and the ERC. We obtained written informed consent from the caregivers of the children before conducting the interviews.

## Results

### Demographic and clinical characteristics

Among the severe pneumonia cases, 28% were in the 0-5 months age group, 33% in the 6-11 months age group, 26% in the 12-23 months age group and 13% were in the 24-59 months age group. Only 31.8% completed EPI (expanded programme on immunization) schedule vaccination. Notably, 100% completed BCG vaccination. Common presenting complaints were cough (98.7%), vomit after cough (32.9%), and chills/rigor (14.4%) (Table 1).

### Danger signs and symptoms

Among danger signs and symptoms, the most common was fast breathing (99.4%). Of these, 68.6% experienced fast breathing for a duration of 0 to 3 days. Similarly, a majority of cases (67.8%) had difficulties in breathing lasting 0 to 3 days. Nearly all cases (98.1%) showed signs of chest in-drawing while 33.3% experienced hypoxemia for 0 to 3 hours. A significant proportion (87.0%) of cases experienced respiratory distress, with 75.2% of severe respiratory distress cases lasting 0 to 3 days. Additionally, about 20% of children were unable to breastfeed or drink (Table 2).

### Incidence of severe pneumonia

From 1st June 2018 to October 2019, a total of 12,303 children were included in the study. A total of 154 children suspected a physician examined severe pneumonia at the primary health center to obtain a clinical diagnosis. The overall severe pneumonia incidence was 1.3% with the incidence being higher in the Dhaka district at 2%, while it was 1.1% in the Kishoregonj district (data not shown in tables or figures). Specifically, within the Dhaka district, the incidence of severe pneumonia was 2% in Dhamrai sub-district. In contrast, within the Kishoregonj district, the incidence was 0.1% in Karimganj, 1.2% in Pakundia and 0.9% in Sadar sub-districts (Table 3).

### Health care-seeking behavior

The majority (44.2%) of the mothers or caregivers sought health care services for their children in a nearby pharmacy (drugstore) or local health care provider. Approximately one in four mothers or caregivers visited district hospitals and private clinics. About 37% received health care services from HFWCs, 22.1% in the health complexes, and only 3.2% in the community clinics (Table 4).

**Table 1. Distribution of severe Pneumonia by demographic and clinical characteristics.**

| Indicator | Male (n=99) n (%) | Female (n=55) n (%) | Total (n=154) n (%) |
|---|---|---|---|
| **Age categories** | | | |
| 0-5 months | 28 (28.3) | 15 (27.3) | 43 (27.9) |
| 6-11months | 36 (36.4) | 15 (27.3) | 51 (33.1) |
| 12-23 months | 21 (21.2) | 19 (34.5) | 40 (26.0) |
| 24-59 months | 14 (14.1) | 6 (10.9) | 20 (13.0) |
| **Current EPI status of the child** | | | |
| Complete (EPI schedule) | 31 (31.3) | 18 (32.7) | 49 (31.8) |
| Running | 58 (58.6) | 30 (54.5) | 88 (57.1) |
| Incomplete | 10 (10.1) | 7 (12.7) | 17 (11.0) |
| **Receipt of the vaccines** | | | |
| BCG | 99 (100.0) | 55 (100.0) | 154 (100.0) |
| Pentavalent (1st dose) | 99 (100.0) | 54 (98.2) | 153 (99.4) |
| Pentavalent (2nd dose) | 97 (98.0) | 53 (96.4) | 150 (97.4) |
| Pentavalent (3rd dose) | 96 (97.0) | 50 (90.9) | 146 (94.8) |
| **Duration of child's illness before reporting** | | | |
| 0 – 3 days | 76 (76.8) | 39 (70.9) | 115 (74.7) |
| 4 – 5 days | 15 (15.2) | 13 (23.6) | 28 (18.2) |
| ≥6 days | 8 (8.1) | 3 (5.5) | 11 (7.1) |
| **Duration of child's fever** | | | |
| 0 – 3 days | 54 (72.0) | 37 (74.0) | 91 (72.8) |
| 4 – 5 days | 12 (16.0) | 9 (18.0) | 21 (16.8) |
| ≥6 days | 9 (12.0) | 4 (8.0) | 13 (10.4) |
| **Severe fever** | | | |
| Yes | 46 (61.3) | 32 (64.0) | 78 (62.4) |
| **Continuous fever** | | | |
| Continuous | 33 (44.0) | 32 (64.0) | 65 (52.0) |
| On and off | 42 (56.0) | 18 (36.0) | 60 (48.0) |
| **Children had chills/rigor** | | | |
| Yes | 12 (16.0) | 6 (12.0) | 18 (14.4) |
| **Children had cough** | | | |
| Yes | 98 (99.0) | 54 (98.2) | 152 (98.7) |
| **Duration of cough** | | | |
| 0 – 3 days | 44 (44.9) | 20 (37.0) | 64 (42.1) |
| 4 – 5 days | 29 (29.6) | 21 (38.9) | 50 (32.9) |
| ≥6 days | 25 (25.5) | 13 (24.1) | 38 (25.0) |
| **Severe cough** | | | |
| Yes | 68 (69.4) | 41 (75.9) | 109 (71.7) |
| **Child vomit after cough** | | | |
| Yes | 34 (34.7) | 16 (29.6) | 50 (32.9) |

## Barriers to seeking healthcare

Approximately 25% (38) of caregivers reported facing barriers to accessing healthcare (Fig 2). Among those who reported barriers, 39.5% cited an inability to find transportation as the main constraint, followed by 26.3% who named the high cost of treatment, 10.5% who faced natural calamities for example floods, and 2.6% who encountered political instability for example strike (Fig 3).

Table 2. Distribution of danger signs and symptoms.

| Indicator | Male (n=99) n (%) | Female (n=55) n (%) | Total (n=154) n (%) |
|---|---|---|---|
| **The child had fast breathing** | | | |
| Yes | 99 (100.0) | 54 (100.0) | 153 (99.4) |
| **Duration of child's fast breathing** | | | |
| 0 – 3 days | 63 (63.6) | 42 (77.8) | 105 (68.6) |
| 4 – 5 days | 26 (26.3) | 8 (14.8) | 34 (22.2) |
| ≥6 days | 10 (10.1) | 4 (7.4) | 14 (9.2) |
| **Duration of the child's difficulty in breathing** | | | |
| 0 – 3 days | 63 (64.3) | 40 (74.1) | 103 (67.8) |
| 4 – 5 days | 24 (24.5) | 11 (20.4) | 35 (23.0) |
| ≥6 days | 11 (11.2) | 3 (5.6) | 14 (9.2) |
| **Children had chest in-drawing** | | | |
| Yes | 97 (98.0) | 54 (98.2) | 151 (98.1) |
| **Duration of child's chest in-drawing** | | | |
| 0 – 3 days | 72 (74.2) | 44 (81.5) | 116 (76.8) |
| 4 – 5 days | 18 (18.6) | 8 (14.8) | 26 (17.2) |
| ≥6 days | 7 (7.2) | 2 (3.7) | 9 (6.0) |
| **Children had respiratory distress** | | | |
| Yes | 90 (90.9) | 44 (80.0) | 134 (87.0) |
| **Duration of severe respiratory distress** | | | |
| 0 – 3 days | 65 (73.0) | 35 (79.5) | 100 (75.2) |
| 4 – 5 days | 13 (14.6) | 7 (15.9) | 20 (15.0) |
| ≥6 days | 11 (12.4) | 2 (4.5) | 13 (9.8) |
| **Children unable to breastfeed or drink** | | | |
| Yes | 17 (17.2) | 13 (23.6) | 30 (19.5) |
| **Children had convulsions** | | | |
| Yes | 2 (2.0) | 0 (0.0) | 2 (1.3) |

Table 3. Incidence rates of childhood severe pneumonia.

| District | Sub-district | *No. of Villages | No. of Bari | No. HH | No. of <5 HH | No. of <5 children | No. of Severe pneumonia | Incidence**, % |
|---|---|---|---|---|---|---|---|---|
| Dhaka | Dhamrai | 28 | 1862 | 7671 | 2560 | 2895 | 59 | 2.0 |
| Kishoreganj | Karimganj | 17 | 606 | 4991 | 2560 | 3125 | 30 | 0.1 |
| | Pakundia | 19 | 513 | 5727 | 2560 | 2995 | 36 | 1.2 |
| | Sadar | 17 | 507 | 5430 | 2560 | 3188 | 29 | 0.9 |
| | **Grand total** | **81** | **3488** | **23819** | **10240** | **12303** | **154** | **1.3** |

*Number of randomly selected villages;

**Multiply the incidence by 100 to express it as a percentage.

## Discussion

We conducted a prospective study in rural Bangladesh to determine the incidence, healthcare-seeking behavior, and barriers related to children and their guardians seeking treatment for childhood severe pneumonia. Key findings were an incidence of severe pneumonia of 1.3% throughout the one-year study period with the majority of the mothers and caregivers using a local health care provider or pharmacy (drugstore) in close proximity for their children's

**Table 4. Access to healthcare services.**

| Indicator* | Male (n=99) n (%) | Female (n=55) n (%) | Total (n=154) n (%) |
|---|---|---|---|
| **The place for availing health care services** | | | |
| District Hospital | 25 (25.3) | 11 (20.0) | 36 (23.4) |
| Health and Family Welfare Centre | 21 (21.2) | 19 (29.1) | 37 (37.0) |
| Private Clinic | 25 (25.3) | 11 (20.0) | 36 (23.4) |
| Health Complex | 23 (23.2) | 11 (20.0) | 34 (22.1) |
| Community clinic | 2 (2.0) | 3 (5.5) | 5 (3.2) |
| Pharmacy/ Local healthcare provider | 43 (43.4) | 25 (45.5) | 68 (44.2) |
| Other | 5 (5.1) | 3 (5.5) | 8 (5.2) |

*Multiple responses were included.

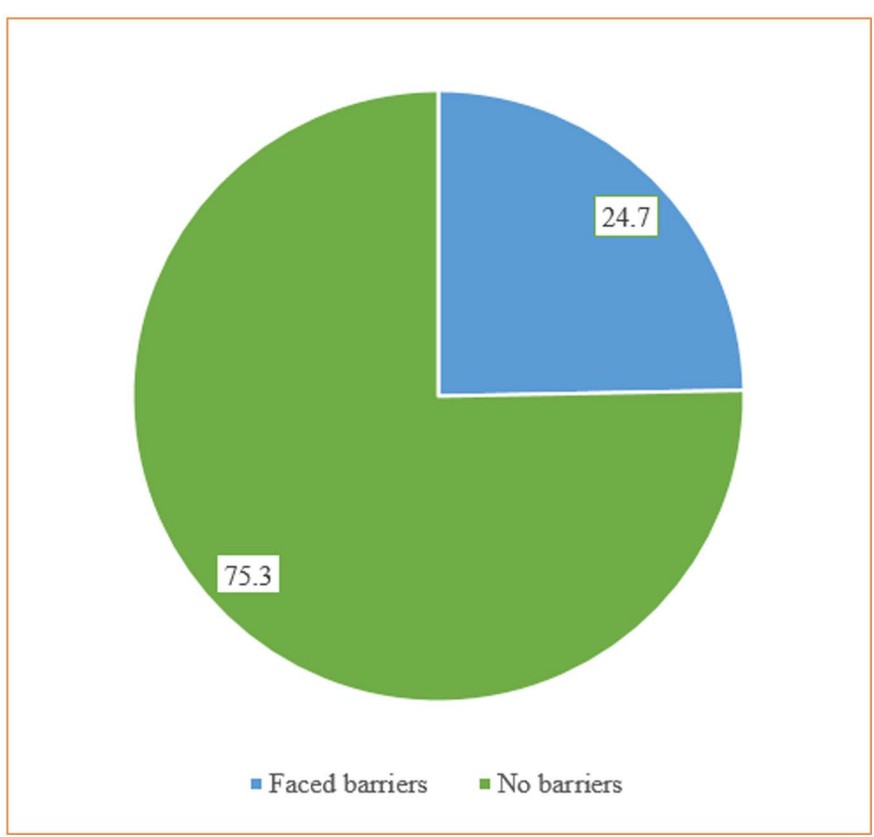

**Fig 2. Distribution of barriers faced or not for accessing healthcare.**

medical needs. About the barriers to accessing health care, one out of four respondents said that they faced barriers. The most notable barriers to care-seeking were the lack of available transportation and the high cost of treatment.

We found the overall severe pneumonia incidence was below 2%. A community-based study in Pakistan aimed to determine the incidence of pneumonia, bacteremia, and invasive pneumococcal disease (IPD) in under-five children reported 1039 clinical cases of pneumonia,

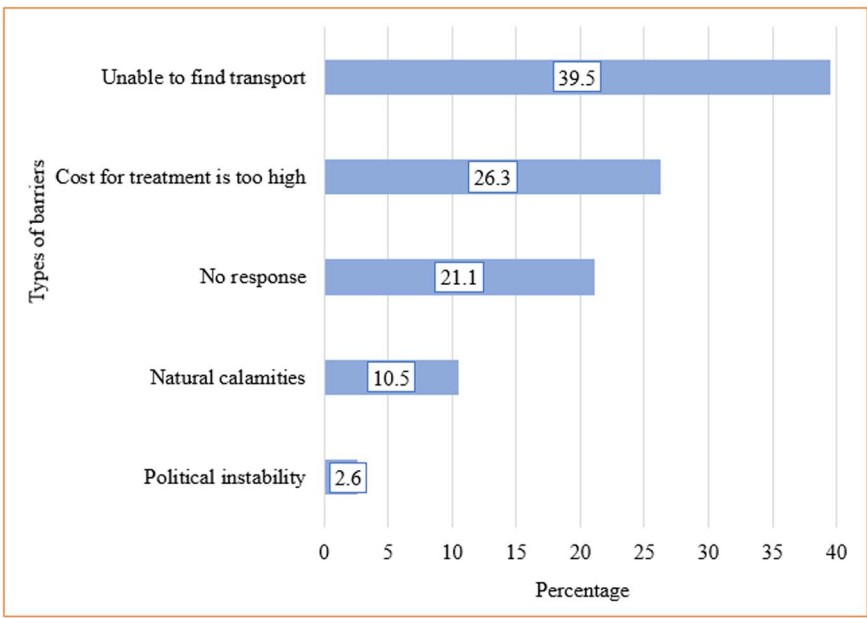

**Fig 3. Types of barriers to accessing healthcare.**

of which 54 were severe pneumonia, and 4 had severe disease (incidence of severe pneumonia in children aged less than one year was defined according to WHO-defined-chest-radiography-positive pneumonia) [31]. Pneumonia caused 12 (22%) deaths among children <5 years old [31]. However, incidence rates of severe pneumonia for other age intervals were estimated through fitting data from a hospital-based multicenter surveillance study. A pathogen was identified from blood of 29 (2.8%) pneumonia cases with the incidence of bacteremia of 912 episodes per 100,000 children–years and a case fatality rate of 8%. The detected IPD incidence was 25 episodes per 100,000 child years and the under-five mortality frequency was 55 per 1000 live births.

The study identified the need to improve care access and increase coverage and equity of pneumonia-preventing vaccines in states with high severe pneumonia burden [32]. Severe pneumonia results in significant morbidity and mortality among Bangladeshi children; vaccines against pneumonia-producing pathogens are crucial for improving child survival in Bangladesh. Since delayed immunization results in lowered protective immunity, vulnerability to pneumonia may increase [33,34]. Completing a pneumococcal vaccination series is necessary for optimal protection against pneumonia or severe pneumonia [35]. Pneumonia or severe pneumonia was noted to be less common in children who had received two or three doses of vaccinations compared to those who received only one dose, supporting to this finding [36]. Delays in vaccination lower population herd immunity and lengthen the window during which a child is susceptible to contracting vaccine-preventable diseases. Children without vaccination are be at higher risk primarily due to increased pneumococcal bacterial transmission [37,38]. According to WHO, immunization remains the sole method to considerably impact the frequency of pneumococcal infections [39]. Starting in the second half of the first year of life, pneumococcal incidence significantly rises as antibody levels fall. Antibodies against Streptococcus pneumonia remain low for up to three years and children between the ages of 3 and 24 months are particularly vulnerable to invasive illnesses brought on by encapsulated bacteria when the level of maternal antibodies decreases, thus it is crucial

to immunize them against pneumococcal infection [40]. The growing resistance to antibiotics highlights the need for vaccination [39]. Children remain susceptible to diseases brought on by Streptococcus pneumoniae if vaccinations, especially the pneumococcal conjugate vaccine (PCV), are delayed [41]. To protect against potentially fatal pediatric diseases, complete vaccination coverage is crucial throughout infancy. Nevertheless, this is insufficient on its own, and vaccinations must be administered by the deadline [42].

Regarding their behavior when seeking healthcare services, the majority of mothers and caregivers preferred to use the nearby pharmacy (drugstore) or local healthcare provider for treatment for their children. This result is consistent with prior research that examined factors influencing appropriate care-seeking in Asia and Africa [43,44]. This study finding might be explained by the close proximity of pharmacy shops near the houses of caregivers, financial costs, as well as underestimation by parents/caregivers of the potential risks of alternative treatments from local healthcare providers [45]. To address this inadequate behavior, the public health resources in the community might consider recruitment and/or training of pharmacists to provide appropriate symptomatic care and to facilitate prompt referrals for severe pneumonia, which has been demonstrated to be beneficial in LMICs [46].

One in four caregivers mentioned difficulties in accessing healthcare services. Among several barriers, the primary ones were those of finances, availability of transportation, and distance to healthcare facilities. Given the effort and expense of transportation required to reach healthcare facilities, mothers' engagement in healthcare-seeking behaviors may be affected by factors such as distance to a healthcare facility [47]. Research has indicated that the WHO's recommendation of a maximum of 5 km as the average distance between one's residence and the closest facility can positively influence the access to healthcare services [48].

Management of childhood severe pneumonia episodes often requires hospitalization, antimicrobial therapy, and supportive care [49]. Treatment of these episodes represents a significant expense, particularly for those requiring hospitalization [49]. Hospital costs for treating childhood pneumonia cases are substantial, particularly in LMICs. This cost has been reported to be United States dollar 48 per outpatient visit for each under-five pneumonia child [50]. This high financial burden prevents families from obtaining medical assistance for pneumonia [51]. These results are consistent with study reports from Ethiopia and Nigeria [52,53]. In Bangladesh, families are responsible for the majority of healthcare costs [54]. Families with low incomes are therefore more likely to receive inadequate care. Children in the poorest Bangladeshi households were observed to be 75% less likely to seek care at a health facility or from a medically trained practitioner for signs of an acute respiratory disease than children living in the wealthiest households [55]. Therefore, to overcome financial obstacles, a steady supply of free medications might be required as well as open and accurate information of fixed facility expenses [56]. In addition, strong policy initiatives to ensure the accessibility and affordability of healthcare services for childhood pneumonia, will be important especially for low-income groups [57].

The result of this study may help refine strategies about case management in resource-limited countries by enabling decisions about the most suitable site of treatment (i.e., home vs. hospital) or the need for additional supportive care. Nation-wide data are needed. There is an urgent need for further research to predict severe pneumonia in the community to address the fundamental knowledge gap to provide optimal management strategies with the potential for significant reductions in morbidity and mortality.

## Conclusion

This study highlights that while the incidence of severe pneumonia in rural areas of Bangladesh is relatively low (below 2%), significant challenges exist to accessing appropriate

healthcare. Pharmacies and local healthcare providers are the primary sources of care, indicating a reliance on easily accessible but potentially less specialized or adequate services. Notable barriers such as the lack of available transportation and high treatment costs hinder effective care-seeking.

Acknowledgmenticddr,b is grateful to the Governments of Bangladesh, and Canada for providing core/unrestricted support. The authors express their gratitude to the study participants. They would also like to acknowledge Gobinda Karmakar for his assistance in developing study sites map.

## Author contributions

**Conceptualization:** Shahjahan Ali, Mohammod Jobayer Chisti, Tahmeed Ahmed, Niklaus Gyr, George J. Fuchs.

**Data curation:** Shahjahan Ali, Tahmeed Ahmed, Niklaus Gyr, George J. Fuchs.

**Formal analysis:** Shahjahan Ali, Md. Tariqujjaman, Niklaus Gyr, George J. Fuchs.

**Investigation:** Shahjahan Ali, Mohammod Jobayer Chisti, Tahmeed Ahmed, Niklaus Gyr.

**Methodology:** Shahjahan Ali, Md. Tariqujjaman, Mohammod Jobayer Chisti, Tahmeed Ahmed, Niklaus Gyr, George J. Fuchs.

**Software:** Shahjahan Ali, Md. Tariqujjaman, Kamal Ibn Amin Chowdhury, Mohammod Jobayer Chisti, Nur H. Alam, Tahmeed Ahmed, Abu S G Faruque, George J. Fuchs.

**Supervision:** Shahjahan Ali, Md. Tariqujjaman, Mohammod Jobayer Chisti, Tahmeed Ahmed, Niklaus Gyr, George J. Fuchs.

**Validation:** Shahjahan Ali, Md. Tariqujjaman, Marufa Sultana, Kamal Ibn Amin Chowdhury, Mohammod Jobayer Chisti, Nur H. Alam, Tahmeed Ahmed, Niklaus Gyr, Abu S G Faruque, George J. Fuchs.

**Visualization:** Shahjahan Ali, Md. Tariqujjaman, Marufa Sultana, Kamal Ibn Amin Chowdhury, Mohammod Jobayer Chisti, Nur H. Alam, Tahmeed Ahmed, Niklaus Gyr, Abu S G Faruque, George J. Fuchs.

**Writing – original draft:** Shahjahan Ali, Md. Tariqujjaman, Arifa Farzana Tanha, Marufa Sultana, Sabiha Nasrin, Kamal Ibn Amin Chowdhury, Mohammod Jobayer Chisti, Nur H. Alam, Tahmeed Ahmed, Niklaus Gyr, Abu S G Faruque, George J. Fuchs.

**Writing – review & editing:** Shahjahan Ali, Md. Tariqujjaman, Arifa Farzana Tanha, Marufa Sultana, Sabiha Nasrin, Kamal Ibn Amin Chowdhury, Mohammod Jobayer Chisti, Nur H. Alam, Tahmeed Ahmed, Niklaus Gyr, Abu S G Faruque, George J. Fuchs.

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
