## [Decision Letter · Decision Letter 0]

6 Mar 2024

PGPH-D-24-00061

Incidence, healthcare-seeking Behavior and Barriers Associated with Seeking Care for Severe Childhood Pneumonia in Rural Bangladesh: A Prospective Study

Dear Dr. Tariqujjaman,

Thank you for submitting your manuscript to PLOS Global Public Health. Firstly, we would like to apologize for the delay in processing your manuscript. It has been exceptionally difficult to secure reviewers to evaluate your study. We have now received one completed review, which is available below. The reviewer has raised significant scientific concerns about the study that need to be addressed in a revision.

Please note that we have only been able to secure a single reviewer to assess your manuscript. We are issuing a decision on your manuscript at this point to prevent further delays in the evaluation of your manuscript. Please be aware that the editor who handles your revised manuscript might find it necessary to invite additional reviewers to assess this work once the revised manuscript is submitted. However, we will aim to proceed on the basis of this single review if possible. 

We look forward to receiving your revised manuscript.

Kind regards,

Miquel Vall-llosera Camps

Staff Editor

Journal Requirements:

Additional Editor Comments (if provided):

Reviewers' comments:

Reviewer's Responses to Questions

**Comments to the Author**

1. Does this manuscript meet PLOS Global Public Health’s publication criteria ? Is the manuscript technically sound, and do the data support the conclusions? The manuscript must describe methodologically and ethically rigorous research with conclusions that are appropriately drawn based on the data presented.

Reviewer #1: Yes

2. Has the statistical analysis been performed appropriately and rigorously?

Reviewer #1: Yes

3. Have the authors made all data underlying the findings in their manuscript fully available (please refer to the Data Availability Statement at the start of the manuscript PDF file)?

Reviewer #1: Yes

4. Is the manuscript presented in an intelligible fashion and written in standard English?

Reviewer #1: Yes

5. Review Comments to the Author

Reviewer #1: Overall, it is a good paper that is showing reduced incidence in pneumonia mortality as well as improved care seeking behaviour in a developing country. However, the authors need to address the following:

1. Line 145, the should remove word "therefore" on the opening of the sentence

2. Lines 203 -207, the authors should clearly state what they were studying "families or children with pneumonia?", the authors should drop the sentence "chest indrawing was the leading complaint, it will confuse readers"

3. Lines 225-226, the percentage of males and females should add to 100% unless otherwise

4. table 3, make the table tidy, remove word indicator and make a good title instead of inserting too much words in the table. there are some confusing figures in the last column of table 3.

5. the authors should discuss the results "result by result" for this to be very clear. it will be interesting to discuss why is the incidence declining, there should be articles especially on pneumococcal vaccine reducing incidence. on the care seeking behaviour, discuss it properly so that they inform the readers why it has reduced.

All the best.

6. PLOS authors have the option to publish the peer review history of their article (what does this mean? ). If published, this will include your full peer review and any attached files.

**Do you want your identity to be public for this peer review?** For information about this choice, including consent withdrawal, please see our Privacy Policy .

Reviewer #1: **Yes: ** Norman Lufesi

---

## [Decision Letter · Decision Letter 1]

23 May 2024

PGPH-D-24-00061R1

Incidence, healthcare-seeking Behavior and Barriers Associated with Seeking Care for Severe Childhood Pneumonia in Rural Bangladesh: A Prospective Study

Dear Dr. Tariqujjaman,

Thank you for submitting your manuscript to PLOS Global Public Health. After careful consideration, we feel that it has merit but does not fully meet PLOS Global Public Health’s publication criteria as it currently stands. Therefore, we invite you to submit a revised version of the manuscript that addresses the points raised during the review process.

Please note that your revision is evaluated by two new reviewers as the previous submission was only assessed by one reviewer.

We look forward to receiving your revised manuscript.

Kind regards,

Jianhong Zhou

Staff Editor

Journal Requirements:

1. Please provide separate figure files in .tif or .eps format only and remove any figures embedded in your manuscript file. Please also ensure all files are under our size limit of 10MB.

Additional Editor Comments (if provided):

Reviewers' comments:

Reviewer's Responses to Questions

**Comments to the Author**

1. If the authors have adequately addressed your comments raised in a previous round of review and you feel that this manuscript is now acceptable for publication, you may indicate that here to bypass the “Comments to the Author” section, enter your conflict of interest statement in the “Confidential to Editor” section, and submit your "Accept" recommendation.

Reviewer #1: All comments have been addressed

Reviewer #2: (No Response)

Reviewer #3: (No Response)

2. Does this manuscript meet PLOS Global Public Health’s publication criteria ? Is the manuscript technically sound, and do the data support the conclusions? The manuscript must describe methodologically and ethically rigorous research with conclusions that are appropriately drawn based on the data presented.

Reviewer #1: Yes

Reviewer #2: Partly

Reviewer #3: No

3. Has the statistical analysis been performed appropriately and rigorously?

Reviewer #1: Yes

Reviewer #2: No

Reviewer #3: N/A

4. Have the authors made all data underlying the findings in their manuscript fully available (please refer to the Data Availability Statement at the start of the manuscript PDF file)?

Reviewer #1: Yes

Reviewer #2: Yes

Reviewer #3: Yes

5. Is the manuscript presented in an intelligible fashion and written in standard English?

Reviewer #1: Yes

Reviewer #2: Yes

Reviewer #3: No

6. Review Comments to the Author

Reviewer #1: (No Response)

Reviewer #2: The study focuses on incidence of severe pneumonia and barriers to access to care for children with severe pneumonia. The survey was conducted at household level.

The authors should consider the following comments to make the manuscript clearer.

The methods section needs to be improved upon to make it easy for the reader to understand and objectively assess the rigor of the study and validity of the results. The authors need to provide clarity on the following:

Study design: The study was nested in a trial, but there is no information about the trial except the link to the site where the trial was registered. A summary with relevant information about the trial and how the current study was nested in there should be provided.

The cases are described as children with severe pneumonia and controls those with non-severe pneumonia. This implies a case-control study design. The authors need to explain how the case-control study was nested in the trial.

In the introduction, the authors indicated that they wanted to collect data on children with severe pneumonia who do not present to the health care settings like health centres and hospitals but remain in the community, to understand the barriers to health seeking. However, in the methods section, it is not clear how these children were identified.

The sampling strategy was stated to be multi-stage but this requires more details. In this survey, how did the team do the sampling from the districts to the households? And how does this relate to the main trial and the case-control study design of the current study?

There is data that was obtained from the health facilities, other sources of care and from households. The authors need to provide clarity on the different data sources and

Clarity on what part of the study was prospective and retrospective will help the reader to gain better understanding of the study results.

The primary aim of the study was to obtain data on the incidence of severe pneumonia. Whereas there is some information on how the data on incidence was obtained (although this requires more information), there is no description on how the data on barriers to care-seeking for children with severe pneumonia was obtained.

There is some information missing on sample size calculation. The formula used is not stated which makes it difficult to assess if the right formula for incidence was used.

The statistical analysis needs improvement. Which data were analysed for the different objectives? What were the outcomes of interest? Which method analysis was used?

Results

The results present information from households about children in general and those with severe pneumonia. There are also results about children seen by the physician. It is difficult to objectively assess these results because the methods used to collect the data that informed the results are not clear.

Results of incidence of severe pneumonia in the control group are presented. However, in the section on study design, the authors indicated that the controls were non-severe pneumonia cases. It is therefore confusing the have severe pneumonia incidence in the control group.

This study seems to have been primarily about incidence of severe pneumonia and barriers to healthcare seeking. The clinical characteristics of the 154 children with severe pneumonia are presented in the tables, disaggregated between males and females. It is not clear how this information relates to the core problem under study (barriers to care-seeking). It is recommended that the data is presented in this context. For example, among those children with cases and controls, how many of them did not seek care, how many delayed to seek care, what were the barriers to seeking care?

Results on barriers to care-seeking are presented but the methods that were used to collect these data were not described.

In the discussion, the authors indicated that the caregivers had no difficulty accessing care, yet at the same time, they show that there were financial barriers. These issues are contradictory. Probably operational definition of access to care and barriers to care would help.

Clarity on the methods and results is required to make an objective assessment of the discussion.

Reviewer #3: The study title explores an important question regarding health seeking behavior and its barriers for severe childhood pneumonia in Bangladesh. The manuscript describes some components but is difficult to follow.

Introduction - has irrelevant text(lines 63-78;82-85;95-108), cites literature10 years or older and does not focus on what is the current status in Bangladesh

Most importantly, how do the authors justify care seeking in a study that is nested in a cluster RCT where participants are expected to report to study clinics with severe pneumonia?

Methods:

How was data on incidence collected is also not clear with unclear information on time of follow up. Especially with regard to additional data on pneumonia reporting to other facilities

Under the heading 'study setting' recruitment procedures are described rather than providing a description of population characteristics and geography of the chosen districts

How was information on barriers to care seeking behavior(third objective) collected is not clear

Sample size calculation is not clear.

Line 193 - presenting 95% CI as inferential measure is unclear

Results: There is a lot of irrelevant data presented in tables and text

Overall severe pneumonia incidence 0.013 (95% CI:0.25-0.28) mentioned here and in the abstract and elsewhere is probably a typo and needs correction.

What is the rationale for presenting severe pneumonia incidence in intervention and control clusters and age categories (Tables 2 and 3)

Table 3, 4 and 6 and description in lines 241-246; 253-256 do not match

Lines 265-280 provide irrelevant text and data that are not part of the objectives

Discussion - very lengthy and difficult to follow.

Lines 304-312 describes delayed/incomplete immunization as a determinant for severe pneumonia but fails to explore why this is so from the parent's perspective, which is an objective that authors are trying to study

Lines 337-346 also does not address the objectives of the manuscript but rather deviates from it.

7. PLOS authors have the option to publish the peer review history of their article (what does this mean? ). If published, this will include your full peer review and any attached files.

**Do you want your identity to be public for this peer review?** For information about this choice, including consent withdrawal, please see our Privacy Policy .

Reviewer #1: No

Reviewer #2: No

Reviewer #3: No

---

## [Decision Letter · Decision Letter 2]

10 Sep 2024

PGPH-D-24-00061R2

Incidence, healthcare-seeking Behavior and Barriers Associated with Seeking Care for Severe Childhood Pneumonia in Rural Bangladesh: A Prospective Study

Dear Dr. Tariqujjaman,

Thank you for submitting your revised manuscript to PLOS Global Public Health. After careful consideration, we feel that it has merit but does not fully meet PLOS Global Public Health’s publication criteria as it currently stands. Therefore, we invite you to submit a revised version of the manuscript that addresses the points raised during the review process.

We look forward to receiving your revised manuscript.

Kind regards,

Priya Rajendran, PhD

Academic Editor

Journal Requirements:

Additional Editor Comments (if provided):

Reviewers' comments:

Reviewer's Responses to Questions

**Comments to the Author**

1. If the authors have adequately addressed your comments raised in a previous round of review and you feel that this manuscript is now acceptable for publication, you may indicate that here to bypass the “Comments to the Author” section, enter your conflict of interest statement in the “Confidential to Editor” section, and submit your "Accept" recommendation.

Reviewer #3: (No Response)

2. Does this manuscript meet PLOS Global Public Health’s publication criteria ? Is the manuscript technically sound, and do the data support the conclusions? The manuscript must describe methodologically and ethically rigorous research with conclusions that are appropriately drawn based on the data presented.

Reviewer #3: No

3. Has the statistical analysis been performed appropriately and rigorously?

Reviewer #3: No

4. Have the authors made all data underlying the findings in their manuscript fully available (please refer to the Data Availability Statement at the start of the manuscript PDF file)?

Reviewer #3: (No Response)

5. Is the manuscript presented in an intelligible fashion and written in standard English?

Reviewer #3: Yes

6. Review Comments to the Author

Reviewer #3: Thank you accepting the suggestions and comments and attempting to address them. Despite this, there are still some significant lacunae that are outlined below

1. The incidence which was calculated and presented correctly (except for the 95% limits) in the first version has now been changed with no reason given why this was done. The current figure although reported as incidence seems more like the prevalence. Similarly the sample size calculations have used prevalence. This needs to be addressed.

2. The three main objectives of this study as reported in the manuscript are

- incidence of severe pneumonia

- care-seeking behavior

- barriers to care seeking

The results if presented according to these objectives makes it easier to understand and read. Presenting results by intervention and control groups makes it very confusing. Likewise, it is difficult to comprehend the distribution of clinical signs based on gender. The table with antibiotics used for treatment of pneumonia is also not part of the study objective and not clear why this has been described.

3. Discussion - this section has been improved but only partially. The second and third paragraph (Lines 312 - 346) has text and findings of studies that are not part of the this study's objectives. While the authors have found some relevant findings related to care seeking and its barriers, discussion regarding the significance of these findings, analyses of why and what could be the ways to overcome them with recommendations is missing.

7. PLOS authors have the option to publish the peer review history of their article (what does this mean? ). If published, this will include your full peer review and any attached files.

**Do you want your identity to be public for this peer review?** For information about this choice, including consent withdrawal, please see our Privacy Policy .

Reviewer #3: No

---

## [Editor Report · Decision Letter 3]

5 Dec 2024

Incidence, healthcare-seeking Behavior and Barriers Associated with Seeking Care for Severe Childhood Pneumonia in Rural Bangladesh: A Prospective Study

PGPH-D-24-00061R3

Dear Tariqujjaman,

We are pleased to inform you that your manuscript 'Incidence, healthcare-seeking Behavior and Barriers Associated with Seeking Care for Severe Childhood Pneumonia in Rural Bangladesh: A Prospective Study' has been provisionally accepted for publication in PLOS Global Public Health.

Best regards,

Priya Rajendran, PhD

Academic Editor
